# Growing Patterns of the *Branca* Chicken Breed—Concentrate vs. Maize-Based Diet

Laura Soares [1,*], Fernando Mata [1], Joaquim L. Cerqueira [1,2] and José Araújo [1,3]

1    CISAS—Center for Research and Development in Agrifood Systems and Sustainability, Instituto Politécnico de Viana do Castelo, Rua Escola Industrial e Comercial Nun'Álvares, 34, 4900-347 Viana do Castelo, Portugal; fernandomata@ipvc.pt (F.M.); cerqueira@esa.ipvc.pt (J.L.C.); pedropi@esa.ipvc.pt (J.A.)
2    Veterinary and Animal Research Centre (CECAV), Quinta de Prados, Apartado 1013, 5001-801 Vila Real, Portugal
3    Mountain Research Centre (CIMO), Instituto Politécnico de Viana do Castelo, Rua D. Mendo Afonso, 147 Refóios do Lima, 4990-706 Ponte de Lima, Portugal
*    Correspondence: laurasoares@esa.ipvc.pt

**Abstract:** Local chicken breeds are threatened with extinction. They must be preserved in order to maintain genetic diversity. The best strategy to preserve these breeds is to understand how they can be made interesting in production systems. With this strategy in mind, this study aimed to understand the growth patterns of the *Branca* breed, which is fed maize and commercial rations. A trial was conducted with $N = 40$ chickens, $n = 10$, in each of the combinations of gender and diet (cocks fed on ration, cocks fed on maize, hens fed on ration, and hens fed on maize). The first step was to determine the best nonlinear model to fit the growth data. After selecting the best fitting model, this was used to estimate the growth, relative growth rate, and instantaneous growth rate curves. The best fit was achieved with the Brody model. Ration-fed cocks grow faster and mature later, as the relative growth rate converges to zero later, while maize-fed hens show slower growth. Maize-fed cocks mature earlier as the relative growth rate converges to zero earlier. Maize-fed cocks and ration-fed hens show intermediate growth patterns compared to ration-fed cocks and maize-fed hens, and similar while comparing with each other. This is a slow-growing breed that reaches the slaughter-ready size at around the fifth month of age.

**Keywords:** *Branca* breed; chicken; cock; growth modeling; hen

## 1. Introduction

Chicken domestication occurred in South and South-east Asia [1] and is currently the most widespread form of livestock globally, owing to the nutritive value of both meat and eggs [2]. The domestication process evolved in response to local resources, needs, climate, and culture [3]. Consequently, numerous breeds have developed worldwide adaptation to diverse biotas, and under the direct influence of human selection [4]. The Portuguese chicken breeds have a long history, however, their standardization occurred only from 2003 for *Pedrês Portuguesa*, *Amarela*, and *Preta* breeds, and more recently in 2010 for the *Branca* breed [5].

The industrialization of animal production systems has resulted in a concentration of production resources, and companies have expanded into multinational sizes. This phenomenon caused the specialization of breeds and specific lines within those breeds, causing many local breeds to lose momentum and fall into decline [6]. In the 1960s, hundreds of breeders coexisted, however, in the 1980s there were only 13 broiler and 12 layer companies responsible for the major industrial production globally. By 2001, there were nine major layer breeding companies, owned by two multinationals (Erich Wesjohann and Hendrix Genetics), and eight broiler breeding companies owned by only four companies (Cobb, Hybro, Hubbard, and Aviagen) [7].

In this context, local poultry genetic resources are endangered, which could be a cause of genetic diversity loss [8]. As a result, many European countries, including Portugal, have implemented policies to favor conservation [9]. Despite the dominance of industrial production systems, numerous small-scale, family production systems persist around the world [7]. In developing countries, local breeds still represent around 95% of the poultry population [7] typically utilized in scavenging production systems, and fed with subproducts and some cereals [10]. Developed countries also maintain family production systems, and commercial subsistence, particularly in Portugal, is based on the belief in the premium quality of these products. These beliefs are rooted in local traditions and gastronomy, and also in organic production and better animal welfare standards [11,12]. As such, it is important to study these local breeds and develop knowledge about their characteristics to enhance local economies based on differentiated niche markets.

*Branca* (Figure 1) is a Portuguese dual-purpose chicken breed, used for eggs and meat, and is still commercially employed in extensive production systems in the northern regions of Entre-Douro and Minho [13]. Soares et al. [14] investigated the growth of three other native Portuguese chicken breeds and concluded that growth is relevant up to 200 days, being almost inexistent after 240 days. These are slow-growing breeds, with performances comparable with other European local breeds raised under small-scale production systems.

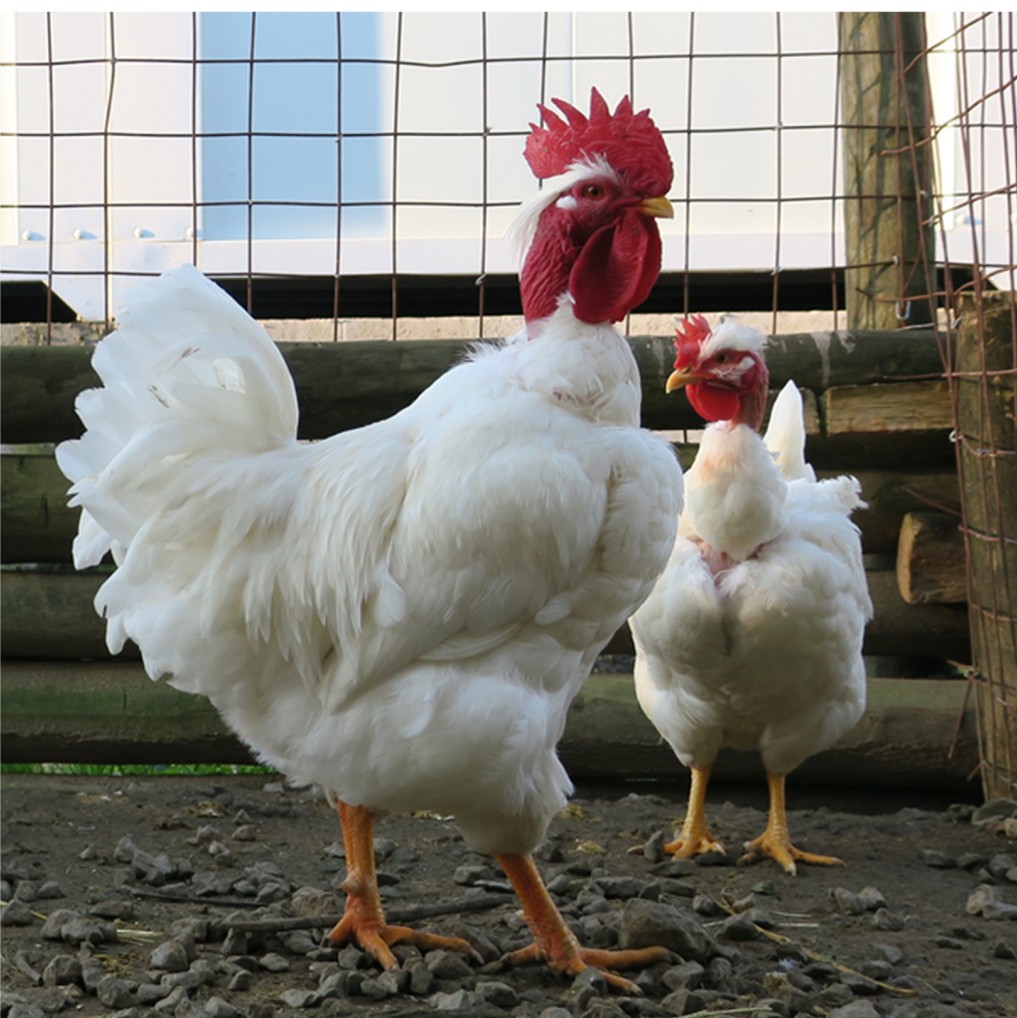

**Figure 1.** *Branca* breed. Cock and hen (Source: authors).

Nonlinear models have been successfully employed to fit the growth curves of animals with their parameterization allowing biological interpretation [15]. These curves are effective in modeling several farm animals including poultry and particularly chicken,

e.g., the studies presented in [9,16,17], describing very well the relationship between age and weight. The objective of this study is to determine the best fit among three models: Logistic, Brody and Gompertz. After the identification of the best-fit model for growth, relative and instantaneous velocity models were also derived. The study was conducted using commercial ration-fed and maize-fed cocks and hens with the ultimate aim of studying the different growth patterns amongst different gendered and feeding standards.

## 2. Materials and Methods

### 2.1. Bird Husbandry and Data Collection

In this trial, we used chicken sourced from AMIBA, Associação de Criadores de Bovinos de Raça Barrosã, the society responsible for the administration of the herd book of several breeds of different species, including cattle sheep and chicken. This society is also responsible for the *Branca* chicken breed herd book.

The trial used $N = 40$ ($n = 20$ cocks, $n = 20$ hens) identified birds induced with a starter. At day 13, $n = 10$ cocks were randomly allocated to the ration-fed group, while the other $n = 10$ cocks were randomly allocated to the maize-fed flocks. The same procedure was followed for the hens. The first flock consisted of 10 cocks and 10 hens and was fed with maize. The second flock, also consisting of 10 cocks and 10 hens, was fed with a standardly designed ration. The two flocks were housed separately. Both flocks were subjected to the same husbandry and stocking density (0.45 m$^2$/beak of covered area, plus 0.9 m$^2$/beak of outside area in a total of 1.35 m$^2$/beak). The chicken coops were timber framed, with a covered roof, and with wire meshed and timber sides. In both flocks, the birds had access to environmentally enriching elements, such as perching devices, sandy soil for dust bathing and nesting facilities. Both feed and water were always available ad libitum. All the birds were weighed on day 13, and weekly thereafter, until day 195 when they were slaughtered.

All the procedures in this study are common husbandry procedures, therefore no ethical considerations deserve committee approval.

### 2.2. Growth Functions Studied

Three nonlinear functions commonly used to fit chicken growth, e.g., [18,19] were used to fit the data: Brody [20], Logistic [21], and Gompertz [22]. The parameterization commonly found for these models is shown in Table 1.

**Table 1.** Parameterizations commonly found for the models used in the present study.

| Model | Equation parameterization |
| --- | --- |
| Brody | $W(t) = a(1 - b \exp(-ct))$ |
| Gompertz | $W(t) = a(\exp(-b \exp(-ct)))$ |
| Logistic | $W(t) = a(1 + b \exp(-ct))^{-1}$ |

Note: $W$ is weight, $t$ is time; $W(t)$ is the weight at time $t$; $a$, $b$, $c$ are the parameters of the equation.

The parameters of the equations shown in Table 1 are associated with the following biological interpretation:

Parameter '$a$' is associated with the mature body weight, the parameter '$c$' is associated with the maturity rate, and the parameter '$b$' is a constant of integration. The maturity rate is associated with growth precocity, representing the rate at which the birds approach their mature weight [23].

### 2.3. Statistical Procedure

The different model parameters were estimated using the Levenberg–Marquardt algorithm, and the least squares method. The adjustment of the curves to the data was made using the NLR (Nonlinear Regression) routine of the software IBM Corp.® SPSS®, Armonk, NY, USA. The version of the software used was the 28.0.1.1 (15). The determination of the best fit was made using the following criteria: the coefficient of determination (r$^2$), the residual mean squares (RMS), and the Mallow's statistic (Cp).

The following seven model assumptions were considered and assessed, as outlined by Frost [24]:

1. The regression model exhibits linearity in both the coefficients and residuals. Guaranteed by the equations of the models.
2. The error terms have a mean of zero. Confirmed with a one-sample *t*-test.
3. The independent variable 'age' does not correlate with the residuals. Verified through Spearman's correlation test.
4. The residuals are not autocorrelated. Verified through the randomness of an ordered residual plot.
5. The residuals do not show heteroscedasticity. Assessed via predicted values versus the residuals plot.
6. Lack of correlation between independent variables. Ensured by the presence of a single independent variable across all models (age).
7. The residuals have a normal distribution. Assessed through a standardized residuals Q-Q plot.

## 3. Results

The three models studied converged and showed a high degree of fitness. The parameter estimates of the models for males and females fed on a ration or maize can be found in Table 2. Table 3 presents the goodness-of-fit statistics, including Mallow's criterion (Cp), the coefficient of determination ($r^2$), and the residual mean square (RMS). As explained in the methodology, prerequisites 1 and 6 are verified directly. The prerequisites 2 and 3 were checked and are also presented in Table 3. The prerequisites 4, 5, and 7 were assessed through the plots included in Appendix A.

The Mallow's criterion is very stable and similar across the models showing a very good fit. This criterion, when having a value close to the number of predictors plus the constant, shows a good model fitness [25]. As we have only one predictor in the models (age) and no constants, the value "1" is indicative of a good fit, across all the models. Considering the coefficient of determination, Brody was the model showing a better adjustment with an average $r^2 = 0.985$, and also a better RMS with an average of 5410 across the four combinations gender/feed. As such, the Brody model was chosen as the most parsimonious to model growth in the *Branca* chicken breed (males and females, ration, and maize-fed).

**Table 2.** Parameters estimates (a, b, c), standard error (SE), and parameter 95% confidence intervals (CI) of the adjusted equations.

| | Functions | a | SE | 95%CI | b | SE | 95%CI | c | SE | 95%CI |
|---|---|---|---|---|---|---|---|---|---|---|
| Hens ration-fed | Brody | 2596.52 | 130.2 | 2324.01; 2869.03 | 1.109 | 0.034 | 1.037; 1.181 | 0.010 | 0.001 | 0.008; 0.013 |
| | Gompertz | 2251.85 | 93.57 | 2056.01; 2447.69 | 2.903 | 0.310 | 2.254; 3.552 | 0.021 | 0.002 | 0.016; 0.026 |
| | Logistic | 2165.13 | 91.38 | 1973.88; 2356.38 | 7.878 | 1.705 | 4.309; 1.447 | 0.031 | 0.004 | 0.023; 0.039 |
| | **Functions** | **a** | **SE** | **95%CI** | **b** | **SE** | **95%CI** | **c** | **SE** | **95%CI** |
| Cocks ration-fed | Brody | 3873.71 | 240.9 | 3369.57; 4377.85 | 1.215 | 0.053 | 1.103; 1.322 | 0.011 | 0.001 | 0.008; 0.014 |
| | Gompertz | 3229.30 | 27.10 | 3172.58; 3286.02 | 5.240 | 0.223 | 4.773; 5.706 | 0.030 | 0.001 | 0.028; 0.031 |
| | Logistic | 3111.77 | 28.55 | 3052.02; 3171.53 | 24.057 | 2.300 | 19.243; 28.871 | 0.046 | 0.002 | 0.043; 0.050 |
| | **Functions** | **a** | **SE** | **95%CI** | **b** | **SE** | **95%CI** | **c** | **SE** | **95%CI** |
| Hens maize-fed | Brody | 2588.74 | 234.6 | 2090.75; 3079.72 | 1.043 | 0.028 | 0.984; 1.102 | 0.007 | 0.001 | 0.005; 0.010 |
| | Gompertz | 2088.10 | 131.2 | 1813.57; 2362.63 | 2.604 | 0.234 | 2.113; 3.094 | 0.017 | 0.002 | 0.012; 0.021 |
| | Logistic | 1968.27 | 112.53 | 1732.74; 2203.80 | 6.924 | 1.248 | 4.312; 9.535 | 0.025 | 0.003 | 0.018; 0.032 |
| | **Functions** | **a** | **SE** | **95%CI** | **b** | **SE** | **95%CI** | **c** | **SE** | **95%CI** |
| Cocks maize-fed | Brody | 2479.02 | 106.8 | 2255.58; 2702.45 | 1.196 | 0.042 | 1.108; 1.285 | 0.012 | 0.001 | 0.010; 0.015 |
| | Gompertz | 2151.00 | 30.0 | 2088.04; 2213.40 | 4.167 | 0.256 | 3.631; 4.702 | 0.028 | 0.001 | 0.026; 0.031 |
| | Logistic | 2502.33 | 99.97 | 2295.19; 2709.46 | 11.147 | 0.875 | 9.315; 12.979 | 0.002 | 0.0001 | 0.0017; 0.0021 |

**Table 3.** Indicators of the quality of adjustment of the different growth models.

| Males | Ration Fed | | | | | Maize Fed | | | | |
|---|---|---|---|---|---|---|---|---|---|---|
| | RMS | $r^2$ | Cp | E$\overline{x}$ | $\rho$Et | RMS | $r^2$ | Cp | E$\overline{x}$ | $\rho$Et |
| Brody | 2758 | 0.98 | 1 | 0 | 0.039 [NS] | 7302 | 0.99 | 1 | 0 | −0.039 [NS] |
| Gompertz | 3162 | 0.99 | 1 | 0 | −0.134 [NS] | 2593 | 0.99 | 1 | 0 | 0.178 [NS] |
| Logistic | 3862 | 0.99 | 1 | 0 | 0.239 [NS] | 3125 | 0.99 | 1 | 0 | 0.161 [NS] |
| Females | Ration Fed | | | | | Maize Fed | | | | |
| | RMS | $r^2$ | Cp | E$\overline{x}$ | $\rho$Et | RMS | $r^2$ | Cp | E$\overline{x}$ | $\rho$Et |
| Brody | 6265 | 0.99 | 1 | 0 | −0.045 [NS] | 5316 | 0.98 | 1 | 0 | −0.030 [NS] |
| Gompertz | 13,384 | 0.97 | 1 | 0 | 0.141 [NS] | 10,451 | 0.97 | 1 | 0 | 0.095 [NS] |
| Logistic | 21,366 | 0.95 | 1 | 0 | 0.212 [NS] | 15,287 | 0.95 | 1 | 0 | 0.153 [NS] |

RMS—Residual mean square, $r^2$—Coefficient of determination, Cp—Mallow's criterion, E$\overline{x}$—Mean error value, $\rho$Et—Spearman's correlation between 'age' and errors, NS—non-significant.

Using the estimated parameters, the growth functions assume the following form:

$$\text{Hens ration-fed} \qquad W(A) = 2596.52 \cdot (1 - 1.109 \cdot e^{(-0.010 \cdot A)}) \qquad (1)$$

$$\text{Cocks ration-fed} \qquad W(A) = 3873.71 \cdot (1 - 1.215 \cdot e^{(-0.011 \cdot A)}) \qquad (2)$$

$$\text{Hens maize-fed} \qquad W(A) = 2588.74 \cdot (1 - 1.043 \cdot e^{(-0.070 \cdot A)}) \qquad (3)$$

$$\text{Cocks maize-fed} \qquad W(A) = 2479.02 \cdot (1 - 1.196 \cdot e^{(-0.012 \cdot A)}) \qquad (4)$$

where $W$ is the weight (g) at age $A$ (days). These growth curves are represented in Figure 2A.

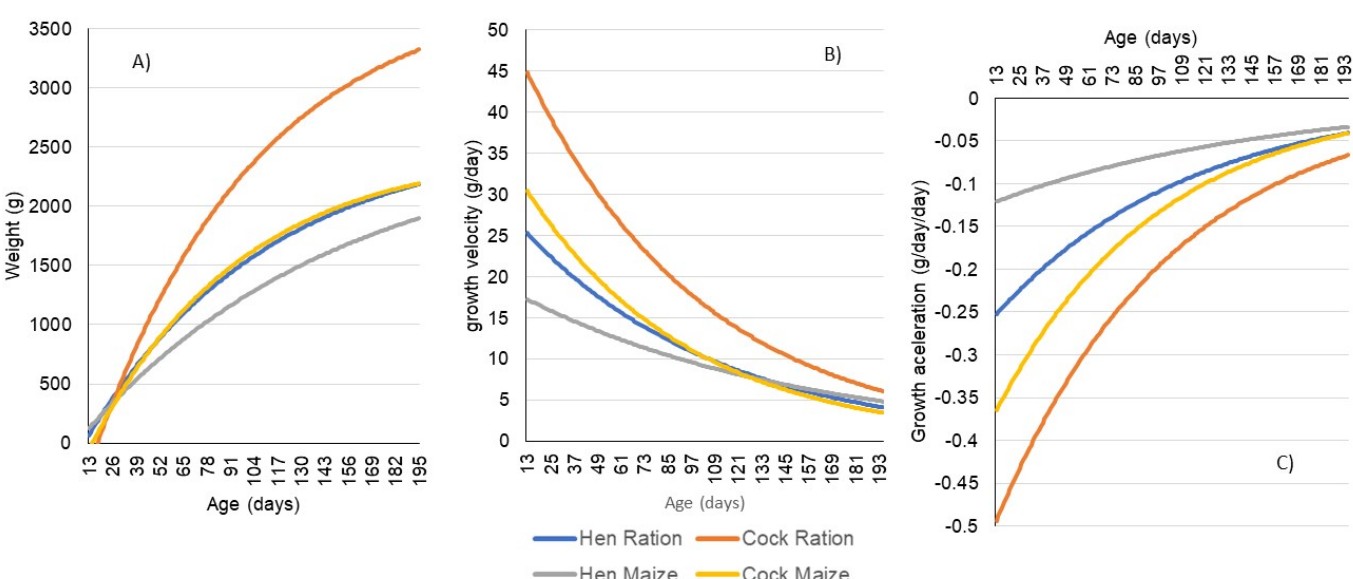

**Figure 2.** Models explaining the growths of hens and cocks fed on ration or maize. (**A**)—growth, (**B**)—relative growth rate (growth velocity), (**C**)—instantaneous growth rate (growth acceleration).

The first derivative functions (Equations (5)–(8)) are used to represent the relative growth rate through age (growth velocity) (Figure 2B) and the second derivative functions (Equations (9)–(12)) represents the instantaneous growth rate through time (growth acceleration) (Figure 2C).

Relative growth rate functions:

$$\text{Hens ration-fed} \qquad \frac{dW(A)}{dA} = 28.7954068 \cdot e^{(-0.010 \cdot A)} \qquad (5)$$

Cocks ration-fed　　　$\dfrac{dW(A)}{dA} = 51.77213415 \cdot e^{(-0.011 \cdot A)}$　　　　(6)

Hens maize-fed　　　$\dfrac{dW(A)}{dA} = 18.90039074 \cdot e^{(-0.007 \cdot A)}$　　　　(7)

Cocks maize-fed　　　$\dfrac{dW(A)}{dA} = 35.57889504 \cdot e^{(-0.012 \cdot A)}$　　　　(8)

Instantaneous growth rate functions:

Hensr ation-fed　　　$\dfrac{d^2 W(A)}{dA} = -0.28795406 \cdot e^{(-0.010 \cdot A)}$　　　　(9)

Cocks ration-fed　　　$\dfrac{d^2 W(A)}{dA} = -0.56949 \cdot e^{(-0.011 \cdot A)}$　　　　(10)

Hens maize-fed　　　$\dfrac{d^2 W(A)}{dA} = -0.1323 \cdot e^{(-0.007A)}$　　　　(11)

Cocks maize-fed　　　$\dfrac{d^2 W(A)}{dA} = -0.42694 \cdot e^{(-0.012 \cdot A)}$　　　　(12)

Considering Figure 2, it is evident that cocks achieve higher weights compared to hens, and birds fed on ration also grow to higher weights. However, the growth curves of cocks fed on maize and hens fed on ration coincide, revealing very similar growth patterns. As cocks reach higher weights, they also grow rapidly, resulting in a higher growth velocity. Hens fed on maize exhibit a flatter growth velocity; initially, they grow slowly, however, around day 150 their velocity surpasses that of hens fed on ration and cocks fed on maize. A similar trend is observed between hens fed on ration and cocks fed on maize. Regarding growth acceleration, we can observe that all four curves decrease (negative increase) the acceleration as growth progresses aligning with the observed decline in velocity. The acceleration decreases more rapidly in birds with faster growth and higher final weights.

## 4. Discussion

The results obtained indicate that ration-fed cocks exhibit faster growth and mature later, as the relative growth rate converges to zero at a later stage, whereas maize-fed hens have a slower growth. Maize-fed cocks mature earlier as the relative growth rate converges to zero earlier. Maize-fed cocks and ration-fed hens have intermediate growing patterns when compared with the previous groups, and similar patterns when compared with each other.

Concerning the live weight on day 195, ration-fed cocks reach 3323 g, while maize-fed cocks reach 2193 g, similar to ration-fed hens (2187 g) and followed by maize-fed hens (1899 g). Meira et al.'s [13] results on ration-fed cocks slaughtered on average at day 280 (38 to 40 weeks) with 3484 g and hens slaughtered on average at day 805 (110 to 120 weeks) with 2518 g indicate that growth after day 195 is insignificant within the time required to reach the weight. The traditional slaughtering age of hens tends to be longer to allow at least two laying seasons, as this is a dual-purpose breed. Similar results were obtained by Meira et al. [12] while comparing the four autochthon Portuguese breeds, with the *Branca* breed cocks weighing 3.5 kg averaging again aged around 280 days (38 to 40 weeks). The *Branca* breed was identified as the heaviest in this comparative study which tallies with the results obtained by Brito et al. [25], considering *Branca* to be "the heaviest, largest and biggest in shank diameter".

The weight differences between genders are expected, are common in chicken breeds, and are associated with the influence generated by sexual hormones in the metabolic processes of lipids and glucose observed in many species [26]. More recently it has also been

rereported that sexual differences in growth patterns of chicken broilers are also associated with the different microbiology found in the caecum of cocks and hens [27,28]. These differences are related to the different microbiota capacities in different genders. While cocks' caecal microbiota is more directed to glycan metabolism, hens' caecal microbiota is more directed to lipid metabolism [28].

The differences observed between feeding programs (ration vs. maize) are associated with a lack of balance in the diet, especially with the protein deficit observed in maize. While a commercial ration may contain around 20% of protein, maize contains around 9% only. Deficiencies in minerals, vitamins and essential amino acids may also be considered. Another important factor lies in the presence of coccidiostatic agents in commercial rations, responsible for the regulation of coccidia in the gut of poultry, allowing higher levels of nutrient absorption by the birds [29].

It is worth noting however, that *Branca* chickens are normally used in extensive, semi-scavenging production systems with access to land where they can forage for food, which normally includes a diversity of invertebrates, and other by-products [12,13] contributing, therefore, to a higher balance of protein in the birds' diet.

The ideal slaughtering age in these slow-growing breeds is earlier than 195 days, as the increments from 150 days forward are limited. These are results also obtained in other slow-growing breeds such as *Milanino* and *Padovana* [30] or *Bianca di Saluzzo* and *Bionda Piemontese* [31], however, other considerations are taken into account in traditional production systems, such as keeping hens for laying or slaughtering only for especial occasions.

This study has some limitations in terms of sample size and sampling points. Since the breed is rare, it is not always possible to find a higher number of birds to enter a trial. The sampling points should also be more frequent, especially in the early days of the life of the birds and up to the end of the first month, as it is in this period that normally an inflection point may be observed in the growth curves. This study captured the weight at day 13 only and weekly after that, which may have contributed to a less reliable curve in the early stages of growth.

Future studies should also direct their attention to comparing different growing patterns between ration-fed birds and traditionally fed birds. As explained before, the traditional production systems have access to foraging areas where a more balanced diet, normally supplemented with maize, may be provided.

## 5. Conclusions

Under the conditions of this study, the Brody function was found the best to fit the growth data of *Branca* breed chicken. Ration-fed chicken had an obviously greater growth rate, as well as cocks in relation to hens. In the future, research should be conducted on the potential for growth in free-range conditions. This can be explored once chickens can supplement their maize feed with other by-products and protein from foraged invertebrates. The ideal slaughtering age is achieved around five months or 150 days of age.

**Author Contributions:** Conceptualization, L.S.; methodology, L.S., F.M., J.A. and J.L.C.; formal analysis, F.M.; investigation, L.S., F.M., J.L.C. and J.A.; resources, L.S., J.L.C. and J.A.; data curation, L.S.; writing—original draft preparation, F.M.; writing—review and editing, L.S., F.M., J.A. and J.L.C.; supervision, L.S.; project administration, L.S.; funding acquisition, L.S. All authors have read and agreed to the published version of the manuscript.

**Funding:** This research was funded by the Instituto Politécnico de Viana do Castelo, grant number: BI_01_2021_Raça Branca. The APC was funded by the CISAS—Center for Research and Development in Agrifood Systems and Sustainability.

**Institutional Review Board Statement:** Not applicable.

**Data Availability Statement:** The data presented in this study are available on reasonable request from the corresponding author.

**Acknowledgments:** To the Foundation for Science and Technology (FCT, Portugal) for financial support to CISAS UIDB/05937/2020 and UIDP/05937/2020, including the contract of F. Mata. To AMIBA—Associação de Criadores de Bovinos de Raça Barrosã for donating the birds used in this study.

**Conflicts of Interest:** The authors declare no conflict of interest.

## Appendix A

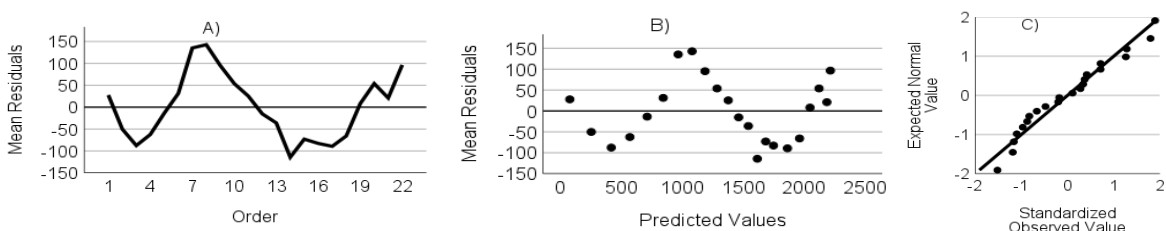

**Figure A1.** Brody model for female chickens ration-fed. (**A**) ordered residual plot, (**B**) residuals versus predicted value plot, (**C**) standardized residuals Q-Q plot.

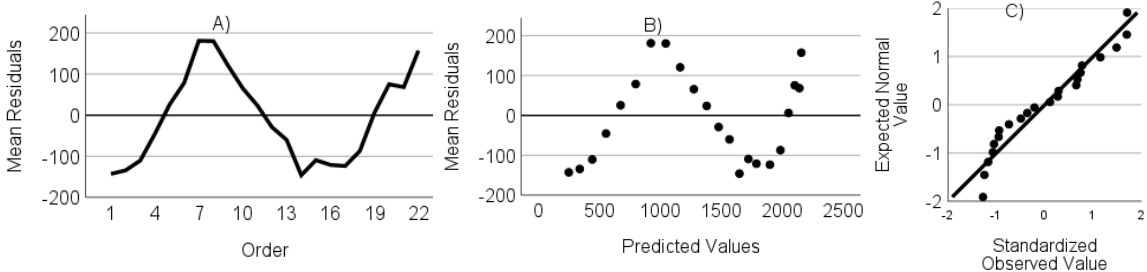

**Figure A2.** Brody model for male chickens ration-fed. (**A**) ordered residual plot, (**B**) residuals versus predicted value plot, (**C**) standardized residuals Q-Q plot.

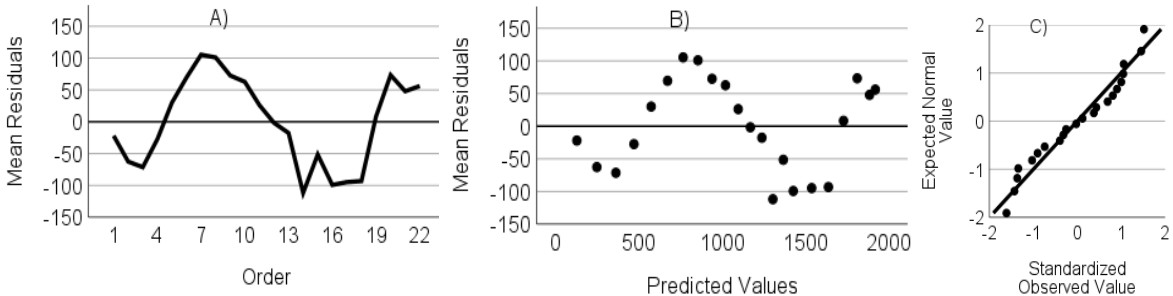

**Figure A3.** Brody model for female chickens maize-fed. (**A**) ordered residual plot, (**B**) residuals versus predicted value plot, (**C**) standardized residuals Q-Q plot.

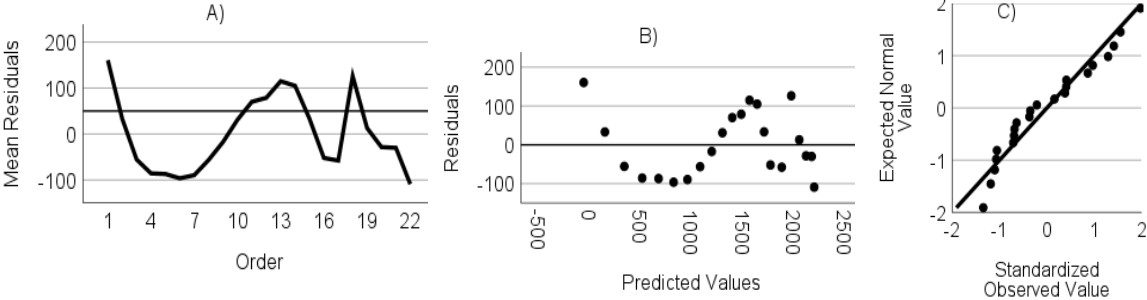

**Figure A4.** Brody model for male chickens maize-fed. (**A**) ordered residual plot, (**B**) residuals versus predicted value plot, (**C**) standardized residuals Q-Q plot.

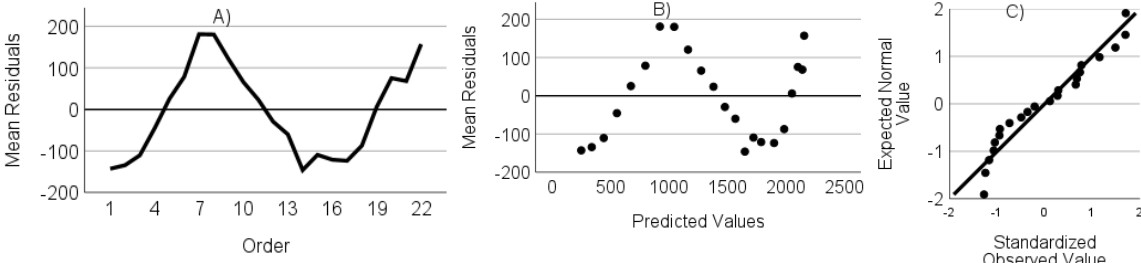

**Figure A5.** Gompertz model for female chickens ration-fed. (**A**) ordered residual plot, (**B**) residuals versus predicted value plot, (**C**) standardized residuals Q-Q plot.

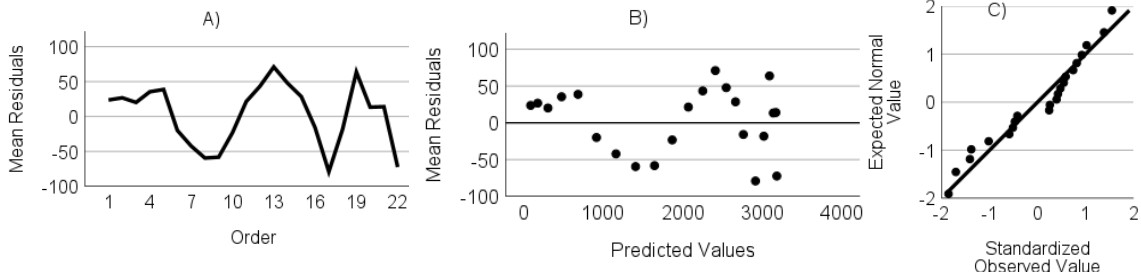

**Figure A6.** Gompertz model for male chickens ration-fed. (**A**) ordered residual plot, (**B**) residuals versus predicted value plot, (**C**) standardized residuals Q-Q plot.

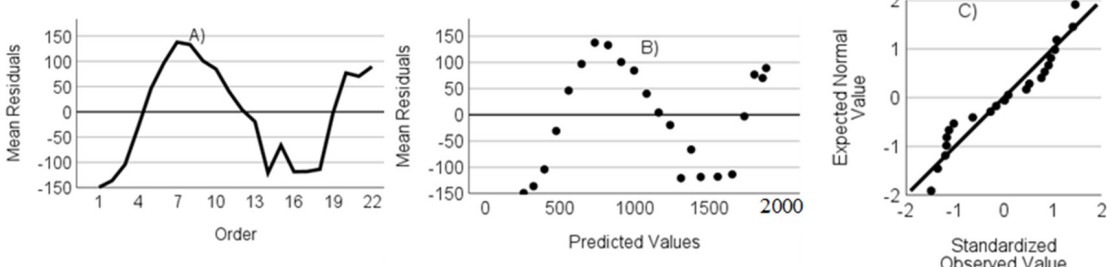

**Figure A7.** Gompertz model for female chickens maize-fed. (**A**) ordered residual plot, (**B**) residuals versus predicted value plot, (**C**) standardized residuals Q-Q plot.

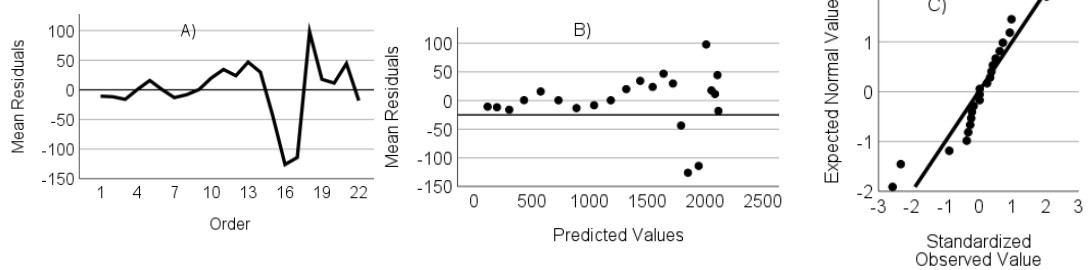

**Figure A8.** Gompertz model for male chickens maize-fed. (**A**) ordered residual plot, (**B**) residuals versus predicted value plot, (**C**) standardized residuals Q-Q plot.

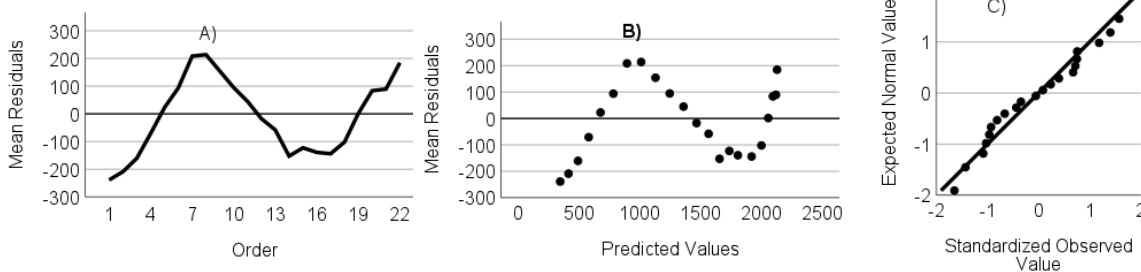

**Figure A9.** Logistic model for female chickens ration-fed. (**A**) ordered residual plot, (**B**) residuals versus predicted value plot, (**C**) standardized residuals Q-Q plot.

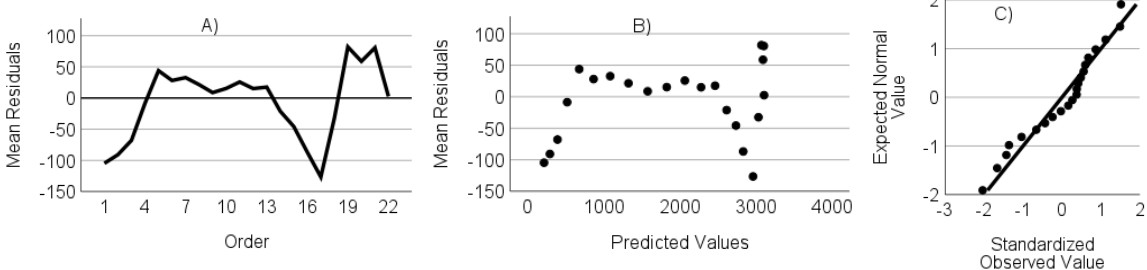

**Figure A10.** Logistic model for male chickens ration-fed. (**A**) ordered residual plot, (**B**) residuals versus predicted value plot, (**C**) standardized residuals Q-Q plot.

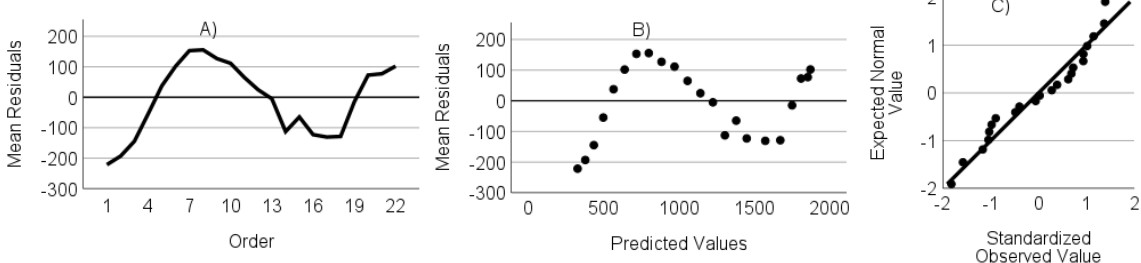

**Figure A11.** Logistic model for female chickens maize-fed. (**A**) ordered residual plot, (**B**) residuals versus predicted value plot, (**C**) standardized residuals Q-Q plot.

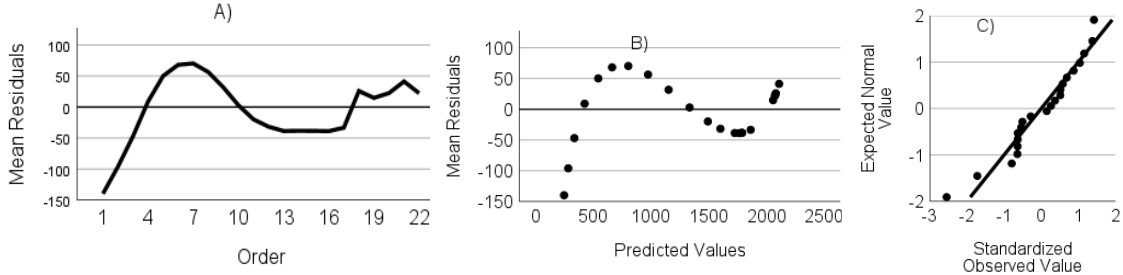

**Figure A12.** Logistic model for male chickens maize-fed. (**A**) ordered residual plot, (**B**) residuals versus predicted value plot, (**C**) standardized residuals Q-Q plot.

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
