# Peer review of "Growing Patterns of the Branca Chicken Breed—Concentrate vs. Maize-Based Diet"

_agriculture, doi:10.3390/agriculture13122282_

Round 1
Reviewer 1 Report
Comments and Suggestions for Authors
Authors are requested to include more information on result interpretation. For instance, in Row 140, you mentioned, 'Brody was the model showing a better adjustment with an average r2=0.985, and also a better RMS with an average of 5410,' but there's no indication of the variables it's compared to.
Though results are presented in Figure 1, the data remains uninterpreted. Does the model used provide P-values? If so, consider reporting them. If not, provide further explanation on how you determined one variable is statistically greater than others.
Authors are requested to revise the manuscript for a smooth flow of information, making it easier to follow.
For examples:
16: We designed atrial using N=40 chicken, 10 in each of the combinations of gender and diet.
22-25: Maize-fed cocks and ration-fed hens have intermediate growing patterns while comparing with the previous groups, and similar while comparing with each other. This is a slow-growing breed that reaches the slaughtering mature size around the fifth month of age.
79: At day 13, 10 of each cock and hen were randomly allocated into two different flocks.
204: The weight differences between genders are expected, are common in chicken breeds
243-246: In the future growth in free-range conditions should be investigated, once chickens can complement their maize feed with other sub-products and the protein from invertebrates foraged. The ideal slaughtering age is achieved around five months or 150 days of age.
Authors are requested to revise the manuscript for a smooth flow of information, making it easier to follow.
For examples:
16: We designed atrial using N=40 chicken, 10 in each of the combinations of gender and diet.
22-25: Maize-fed cocks and ration-fed hens have intermediate growing patterns while comparing with the previous groups, and similar while comparing with each other. This is a slow-growing breed that reaches the slaughtering mature size around the fifth month of age.
79: At day 13, 10 of each cock and hen were randomly allocated into two different flocks.
204: The weight differences between genders are expected, are common in chicken breeds
243-246: In the future growth in free-range conditions should be investigated, once chickens can complement their maize feed with other sub-products and the protein from invertebrates foraged. The ideal slaughtering age is achieved around five months or 150 days of age.
Author Response
Reviewer 1
Dear reviewer 1 thank you very much for your feedback and for helping us to improve this manuscript
Authors are requested to include more information on result interpretation. For instance, in Row 140, you mentioned, 'Brody was the model showing a better adjustment with an average r2=0.985, and also a better RMS with an average of 5410,' but there's no indication of the variables it's compared to.
Being averages (means) we are referring to the 4 combinations of gender and feed both for R2 and RMS. We have now added additional information to complete the sentence more clearly.
Though results are presented in Figure 1, the data remains uninterpreted. Does the model used provide P-values? If so, consider reporting them. If not, provide further explanation on how you determined one variable is statistically greater than others.
The adjustment of a curve does not provide p-values. It produces however the stats reported to allow the evaluation of the degree of adjustment.
The following paragraph has now been added to the results section:
In consideration to Figure 1, we can observe that cocks grow to higher weights in comparison to hens, and birds fed on ration also grow to higher weights. However, the growth curves of cocks fed on maize and hens fed on ration are coincident, revealing very similar growth. As cocks grow to higher weights, they also grow quickly and, therefore, their growth velocity is higher. Hens fed on maize have a flatter growth velocity, initially, they grow slowly, however, from approximately day 150 their velocity overtakes those hens fed on ration and cocks fed on maize. The same phenomenon can be observed between hens fed on ration and cocks fed on maize. Finally, concerning the growth acceleration, we can observe that all four curves decrease (negative increase) the acceleration as growth takes place, and in consonance with the observed decrease in velocity. The acceleration decreases faster in birds growing faster and achieving higher weights.
Authors are requested to revise the manuscript for a smooth flow of information, making it easier to follow.
For examples:
16: We designed atrial using N=40 chicken, 10 in each of the combinations of gender and diet.
Sentence revised
22-25: Maize-fed cocks and ration-fed hens have intermediate growing patterns while comparing with the previous groups, and similar while comparing with each other. This is a slow-growing breed that reaches the slaughtering mature size around the fifth month of age.
Sentence revised
79: At day 13, 10 of each cock and hen were randomly allocated into two different flocks.
Sentence revised
204: The weight differences between genders are expected, are common in chicken breeds
The sentence is correct. Please mind commas,…. are expected…, are common…., and are associated…
243-246: In the future growth in free-range conditions should be investigated, once chickens can complement their maize feed with other sub-products and the protein from invertebrates foraged. The ideal slaughtering age is achieved around five months or 150 days of age.
The sentence was now rephrased
Reviewer 2 Report
Comments and Suggestions for Authors
Please see the attachment.

Author Response
Reviewer 2
Dear reviewer 2 thank you very much for your feedback and for helping us to improve this manuscript
Comments : This paper reports some useful information about the local chicken breed, which is important for meat production.
Thank you
The main problem is that samples of only 10 chickens per treatment were used to build the regression model.
While the number seems low, please note that a total of 40 birds were used in the trial. Please also consider the comments produced in “limitations” in our discussion section, as well as future directions. This is a very rare breed, therefore not easy, at least at the moment, to produce trials with larger numbers.
Further concerns and comments are listed below. 1. There are only 10 replicates in each treatment, which is a very small number of chickens. Is it possible to increase the number of chickens per treatment? Or how to explain that this information is meaningful enough for publication?
Commented above.
- Table 1: Please add the footnote indicating each parameter in the equation, e.g. a= ?, b = ?, t= ?, ct =?
Now added
- The English language is quite difficult to understand, there are many places in the manuscript where corrections are needed. For example: 3.1 Abstract There are some suggestions (in red font) to make the text more understandable, and please clarify or rewrite the places marked in yellow to make them more understandable.
We have now revisited and made amendments throughout
3.2 Line 142: “the Brody model was chosen as the best fit to model the four curves” needs to
be rewritten.
Now rewritten
3.3 Conclusion should be rewritten to make it more understandable, and a correction of
the English is also needed. What is “the protein from invertebrates foraged” in line 245?
Rewritten. Protein from snails and insects foraged by the birds in free-range systems.
Reviewer 3 Report
Comments and Suggestions for Authors
In this study, Branca chickens were used as experimental materials to study the growth pattern of the breed under corn and commercial diet feeding. The results showed that the Brody model was the most suitable nonlinear model for evaluating growth data. The growth rate of quantitative feeding chickens was significantly higher than that of hens, and the growth rate of cocks was also significantly higher than that of hens. The research results of this paper are rich in data and clear in description, but there are some shortcomings. It is suggested to receive them directly after modification.
1. Lines 79-81, the sample size of each experimental group was only 10 chickens. Will the small sample size affect the accuracy of the results?
2. Materials and Methods, it is suggested that the author supplement the source of the experimental chicken flocks.
3. Lines 110-124, it is suggested to explain the selection basis of software and method in detail.
4. The conclusion of the article indicates that the ideal slaughter age is about 150 days. There is no relevant data in the results section to reflect this conclusion. It is suggested that the author describe how to draw this conclusion in detail in the results section.
Comments on the Quality of English LanguageThe English writing quality of this article is good.
Author Response
Reviewer 3
Dear reviewer 3 thank you very much for your feedback and for helping us to improve this manuscript
In this study, Branca chickens were used as experimental materials to study the growth pattern of the breed under corn and commercial diet feeding. The results showed that the Brody model was the most suitable nonlinear model for evaluating growth data. The growth rate of quantitative feeding chickens was significantly higher than that of hens, and the growth rate of cocks was also significantly higher than that of hens. The research results of this paper are rich in data and clear in description, but there are some shortcomings.
Thank you
It is suggested to receive them directly after modification.
- Lines 79-81, the sample size of each experimental group was only 10 chickens. Will the small sample size affect the accuracy of the results?
We have included a limitations paragraph within the discussion where we refer to the issue
- Materials and Methods, it is suggested that the author supplement the source of the experimental chicken flocks.
Thank you very much for reminding us. We have now added the information to the text and also in acknowledgements as the birds were donated.
- Lines 110-124, it is suggested to explain the selection basis of software and method in detail.
No particular reason for the choice of this software, it is the one available in our research centre.
- The conclusion of the article indicates that the ideal slaughter age is about 150 days. There is no relevant data in the results section to reflect this conclusion. It is suggested that the author describe how to draw this conclusion in detail in the results section.
This aspect is previously referred to in the discussion.